# Analysis of the Flammability and the Mechanical and Electrostatic Discharge Properties of Selected Personal Protective Equipment Used in Oxygen-Enriched Atmosphere in a State of Epidemic Emergency

**DOI:** 10.3390/ijerph191811453

**Published:** 2022-09-12

**Authors:** Adriana Dowbysz, Bożena Kukfisz, Dorota Siuta, Mariola Samsonowicz, Andrzej Maranda, Wojciech Kiciński, Wojciech Wróblewski

**Affiliations:** 1Department of Chemistry, Biology and Biotechnology, Bialystok University of Technology, Wiejska 45A Street, 15-351 Bialystok, Poland; 2Faculty of Security Engineering and Civil Protection, The Main School of Fire Service, Slowackiego Street 52/54, 01-629 Warsaw, Poland; 3Faculty of Process and Environmental Engineering, Lodz University of Technology, 213 Wólczańska Str., 90-924 Lodz, Poland; 4Lukasiewicz Research Network, Institute of Industrial Organic Chemistry, 6 Annopol Str., 03-236 Warsaw, Poland; 5Institute of Chemistry, Military University of Technology, 2 Kaliskiego Str., 00-908 Warsaw, Poland; 6Internal Security Institute, The Main School of Fire Service, Slowackiego Street 52/54, 01-629 Warsaw, Poland

**Keywords:** personal protective equipment, COVID-19, oxygen-enriched atmosphere, fire behavior, mechanical properties

## Abstract

Numerous fires occurring in hospitals during the COVID-19 pandemic highlighted the dangers of the existence of an oxygen-enriched atmosphere. At oxygen concentrations higher than 21%, fires spread faster and more vigorously; thus, the safety of healthcare workers and patients is significantly reduced. Personal protective equipment (PPE) made mainly from plastics is combustible and directly affects their safety. The aim of this study was to assess its fire safety in an oxygen-enriched atmosphere. The thermodynamic properties, fire, and burning behavior of the selected PPE were studied, as well as its mechanical and electrostatic discharge properties. Cotton and disposable aprons were classified as combustible according to their LOI values of 17.17% and 17.39%, respectively. Conall Health A (23.37%) and B/C (23.51%) aprons and the Prion Guard suit (24.51%) were classified as self-extinguishing. The cone calorimeter test revealed that the cotton apron ignites the fastest (at 10 s), while for the polypropylene PPE, flaming combustion starts between 42 and 60 s. The highest peak heat release rates were observed for the disposable apron (62.70 kW/m^2^), Prion Guard suit (61.57 kW/m^2^), and the cotton apron (62.81 kW/m^2^). The mean CO yields were the lowest for these PPEs. Although the Conall Health A and B/C aprons exhibited lower pHRR values, their toxic CO yield values were the highest. The most durable fabrics of the highest maximum tensile strength were the cotton apron (592.1 N) and the Prion Guard suit (274.5 N), which also exhibited the lowest electrification capability. Both fabrics showed the best abrasion resistance of 40,000 and 38,000 cycles, respectively. The abrasion values of other fabrics were significantly lower. The research revealed that the usage of PPE made from polypropylene in an oxygen-enriched atmosphere may pose a fire risk.

## 1. Introduction

The high level of protection of healthcare workers who play a crucial role in the management and treatment of COVID-19 patients is of extreme importance. In the early stages of the pandemic, when there was no assurance suggesting that transmission of SARS-CoV-2 occurs primarily through airborne droplets, it was not known whether personal protective equipment (PPE) such as the respirators or surgical masks used in other respiratory virus infections, would also be effective against COVID-19. However, advances in research have underscored the effectiveness of such PPE, especially face masks and respirators [1]. Other elements of PPE preventing infection are protective clothing, shoes, gloves, and goggles. It is recommended for healthcare workers dealing with confirmed COVID-19 patients in intensive care units to wear fluid-resistant, long-sleeved gowns, disposable fluid-resistant hoods, full-length plastic aprons, disposable full-face visors, FFP3 respirators, two sets of long cuffed gloves, surgical boots/shoes, and disposable boot covers [2].

The major components of PPE are polymers, including polypropylene, polyurethane, polyacrylonitrile, polystyrene, polycarbonate, polyethylene, and polyester [3]. The aliphatic hydrocarbon structure of polypropylene fabrics, being a major component of the PPE in this study, is the main reason for its high flammability. Spontaneous ignition occurs at 345 °C. Moreover, flaming droplets are formed during burning. A high peak of heat release rate and smoke production rate make polypropylene fabrics a serious fire hazard [4]. 

However, exposure to suspected or confirmed COVID-19 patients is not the only risk for healthcare workers. The increased frequency of fires in hospitals (e.g., in India) during the pandemic also demands attention. The high power demand, increase in flammable materials use in wards, limitation of fire safety requirements, and increased oxygen use constitute factors that increase fire hazards [5]. Numerous fire accidents in hospitals appear to be a direct result of the oxygen-enriched atmosphere caused by the work of multiple oxygen ventilators in enclosed areas [6]. According to a report of the Joint Research Centre (JRC), a service of the European Commission, the number of oxygen-related fires between March 2020 and November 2021 significantly exceeded the number of similar incidents compared to the previous period. A death toll of nearly 400 people from at least 60 fires were reported in various countries around the world [7].

Although oxygen is vital to sustain life, exposure to high concentrations of oxygen for a short duration or to lower concentrations of oxygen for a longer time may lead to the occurrence of oxygen toxicity [8]. The physical properties of oxygen, including absence of color, odor, and taste, as well as non-irritability, increase the potential fire hazard due to the human inability to detect its presence [9]. 

An oxidizer, as one of the elements of the fire triangle, is required to produce a fire. Oxygen concentrations higher than 21% cause a fire to spread faster, increase the temperature of the flame, and decrease the minimum temperature or ignition energy to produce combustion. As a consequence, fires are more vigorous and destructive [6] and may become hard to extinguish [10]. Even fire-resistant materials may burn in an atmosphere containing higher oxygen concentrations. The most common combustible material that directly affects the safety of healthcare workers and patients is clothing, which will burn more vigorously in an oxygen-enriched atmosphere [11].

The oxygen measurements carried out in wards hospitalizing SARS-CoV-2 patients revealed that in some compartments the oxygen concentration exceeds its safe value and may reach up to 25.2%. At an oxygen concentration of 23%, the healthcare workers′ clothing and the patients’ bed linen could ignite and catch fire. A further increase of up to 24% of oxygen requires healthcare workers to wear special fire-resistant personal protective equipment. A significant increase in fire hazard would be observed when oxygen concentration reaches 25%, due to the dynamic burn of hair and clothing [12].

The aim of this research is to study the flammability and the mechanical and electrostatic discharge properties of the selected personal protective equipment elements, including the aprons and suits used by healthcare workers during the COVID-19 pandemic, in order to assess fire safety in an oxygen-enriched atmosphere.

## 2. Materials and Methods

### 2.1. Materials

Fabrics from four types of apron and a protective suit were selected for the study. The compositions of the examined fabrics are presented in Table 1.

### 2.2. Methods

#### 2.2.1. Limiting Oxygen Index (LOI)

The limiting oxygen index (LOI) is defined as the smallest volumetric concentration of oxygen in a mixture of oxygen with nitrogen, introduced at a temperature of 23 °C ± 3 °C, at which combustion of a material is maintained [13,14]. The LOIs of specimens (140 mm × 50 mm) were measured using an oxygen index meter (Fire Testing Technology, East Grinstead, UK) according to ISO 4589-2:2017 Plastics—Determination of burning behavior by oxygen index—Part 2: Ambient-temperature test.

#### 2.2.2. Cone Calorimeter Test 

Fire behavior was assessed using a cone calorimeter (Fire Testing Technology, UK), stimulating forced-fire conditions [15]. The specimens (100 mm × 100 mm) placed in the specimen holder were tested at a constant heat flux of 25 kW/m^2^, according to ISO 5660-1:2015 Reaction-to-fire tests—Heat release, smoke production, and mass loss rate—Part 1: Heat release rate (cone calorimeter method) and smoke production rate (dynamic measurement) [16]. Measurements were conducted in three repetitions.

#### 2.2.3. Differential Scanning Calorimeter Test

A differential scanning calorimeter DSC 3500 Sirius (Netzsch, Germany) was used to examine the thermodynamic properties of materials. All material specimens (6 ± 0.1 mg) were placed in the crucible cell under a nitrogen atmosphere, at a flow rate of 40 mL/min. Additionally, the thermal analysis of the cotton apron was assessed in the 21% oxygen and 100% oxygen atmospheres. The heating rate was kept at 10 °C/min, and the observations were carried out in the temperature range of 18–600 °C, with a temperature measurement accuracy of 0.1 K and enthalpy measurement accuracy of <2%. The DSC measurement data analysis was conducted using Proteus version 6.1.0 (Netzsch, Selb, Germany). Measurements were conducted in three repetitions.

#### 2.2.4. Oxygen Bomb Calorimeter Test

The gross heat of combustion (calorific value) QPCS (MJ/kg), defined as the total heat of combustion measured per unit mass [17], was assessed in an oxygen bomb calorimeter (PRECYZJA-BIT, Bydgoszcz, Poland). The measurements were performed according to ISO 1716:2018 Reaction to fire tests for products—Determination of the gross heat of combustion (calorific value) [18]. Combustion of the specimens (1 g) was carried out in an oxygen atmosphere at a pressure of 2 MPa. Measurements were conducted in two repetitions.

#### 2.2.5. Tensile Test

A tensile testing machine (Instron 2519-107, Norwood, USA) was used to test the tensile properties and determine the value of maximum force at which a large number of yarns break simultaneously [19] and the corresponding elongation at break of the studied fabrics. The test was conducted using the test methodology of ISO 13934-1:2013 Textiles—Tensile properties of fabrics—Part 1: Determination of maximum force and elongation at maximum force using the strip method [20]. The specimens (25 mm × 200 mm) were placed in the machine′s clamps. The distance between the clamps was 100 mm, and the sliding speed of the upper beam (the speed at which the fabric sample was stretched) was 100 mm/min. The specimens were conditioned at 20 °C and 65% relative humidity for 24 h before being tested. Measurements were conducted in three repetitions.

#### 2.2.6. Abrasion Resistance Test

A Martindale apparatus (SDLATLAS M235, Rock Hill, USA) was used to test the abrasion resistance of fabrics using the Martindale method, in accordance with ISO 12947-2:2017 Textiles—Determination of abrasion resistance of fabrics by the Martindale method—Part 2: Determination of specimen breakdown [21]. Abrasion occurs with textiles and is caused by friction, resulting in textile material wear [22]. Specimens of a circular shape with a diameter of 60 mm were placed in the special heads of the Martindale apparatus by loading them with a mass of 795 g. Small discs of wool abradant fabric were then continually rubbed against the fabric in a circular motion, until two yarns broke or when there was a noticeable change in appearance. Observations of the specimens during their abrasion were made every 2000 abrasion cycles. The specimens were conditioned at 20 °C and 65% relative humidity for 24 h before being tested. Measurements were conducted in three repetitions.

#### 2.2.7. Electrostatic Discharge Test

The electrostatic discharge properties, half decay time of charges test, was carried out in accordance with EN 1149-3:2004 Protective clothing—Electrostatic properties—Part 3: Test methods for measurement of charge decay [23]. Specimens of a circular shape with a diameter of 55 mm were placed horizontally in a special clamp a few millimeters above an electrode. A step high voltage (1200 V for 30 μs) applied to the electrode generated an electric field that caused electrification of the specimen by induction. The decrease in the magnitude of the charge created on the sample was monitored over a period of 30 s by measuring the field strength of the electrostatic field produced by the charge. The field strength was measured using a non-contact method with a probe placed over the specimen. The specimens were conditioned at 23 °C and 40% relative humidity for 24 h before being tested. The tests were performed under the same conditions. Measurements were conducted in three repetitions.

All measurements were conducted with numerous repetitions to ensure that the obtained results were reliable and repeatable.

## 3. Results and Discussion

### 3.1. Burning Behavior

Despite its simplicity, the LOI method still remains one of the most important screening methods for assessing the burning behavior of various materials [24]. Table 2 shows the results of determining the LOI of the studied fabrics and their classification.

Fabrics exhibiting LOI values of less than 21% are classified as combustible. They burn readily because atmospheric oxygen concentration is sufficient for supporting the burning process [25]. For most fabrics, the lower the LOI value, the more easily the material will burn [26]. Cotton and disposable aprons were classified as combustible according to their LOI values, while Conall Health aprons and the Prion Guard suit were classified as self-extinguishing. The higher LOI values of the latter indicate its lower flammability.

### 3.2. Fire Behavior

The averaged test results obtained from the cone calorimeter during the combustion of fabrics at an external heat flux of 25 kW/m^2^ are presented in Table 3.

Time to ignition (TTI), which defines how quickly a material will ignite under exposure to a certain heat flux [27], varies for the tested materials. The cotton apron has the lowest TTI value of 10 s. A significantly longer time (40 s) is needed for ignition of the Prion Guard suit. Other fabrics exhibit an even higher TTI, with a Conall Health B/C apron being the most difficult to ignite.

The heat release rate (HRR), defined as the amount of heat energy evolved by a material when burned, expressed per unit, is one of the most important characteristics of fire behavior [28]. The heat release rate as a function of time for all studied fabrics is presented in Figure 1. The total duration of the experiment was 1200 s. 

The HRR for all fabrics reached its peak (pHRR) quickly after ignition, and then abruptly dropped. The pHRR values were highest for the cotton (62.81 kW/m^2^) and disposable aprons (62.70 kW/m^2^), as well as for the Prion Guard suit (61.57 kW/m^2^). More than 60% lower values were observed for the Conall Health A (23.65 kW/m^2^) and B/C (18.56 kW/m^2^) aprons. The lowest HRR value obtained for the Conall Health B/C apron may be ascribed to the presence of an additional protective layer against microbiological agents. Compounds used for the manufacture of fabrics with an antimicrobial effect often contain halogen atoms, including chlorine, bromine, or iodine, as well as nitrogen and zinc atoms, which are known for their flame retardant activity [29].The time to pHRR (t_pHRR_), which indicates when the decomposition occurs at its fastest rate [27], increased in line with the increase in TTI for all fabrics. 

The total heat release (THR) curves for all tested fabrics are presented in Figure 2. The highest values were obtained for fabrics exhibiting the highest HRR, i.e., the cotton apron (1.1 MJ/m^2^), Prion Guard suit (1.0 MJ/m^2^), and disposable apron (0.9 MJ/m^2^), respectively. Significantly lower values of 0.3 and 0.2 MJ/m^2^ were obtained for the Conall Health A and B/C aprons, respectively. The THR curves follow a similar pattern for those fabrics that do not contain an inner layer against microbiological agents (with two growth steps clearly marked). In comparison, the THR curve for the Conall Health B/C apron differs from the others by the existence of three growth stages. However, surprisingly, it is not visible on the HRR curve (Figure 1). 

The maximum average rate of heat emission (MARHE), defined as the average energy value generated during each combustion period [30], may be used as a scale of the flammability of the sample [31]. The cotton apron and the Prion Guard suit exhibited the highest MARHE values of 26.4 kW/m^2^ and 13.75 kW/m^2^, respectively. Considerably lower values were obtained for the disposable (9.90 kW/m^2^), Conall Health A (3.59 kW/m^2^), and B/C (2.19 kW/m^2^) aprons.

The toxic carbon monoxide yield is lowest for the cotton apron (132.82 kg/kg) and Prion Guard suit (332.65 kg/kg). Significantly higher values of 562.46 kg/kg, 687.51 kg/kg, and 700.43 kg/kg were obtained for the disposable, Conall Health A, and B/C aprons, respectively. A similar trend of increasing CO_2_ yield was observed, but the highest value was obtained for the Conall Health A apron (253.80 kg/kg).

### 3.3. Thermal Analysis

Figure 3 shows the heating DSC curves of the tested materials. The endothermic thermal effects were observed for each tested fabric. The melting of the disposable apron started at the lowest temperature (i.e., 155.1 °C) among all the tested fabrics and ended at 168.7 °C, with an enthalpy of −42 J/g. The Conall Health A and B/C aprons started melting at slightly higher temperatures of 155.3 °C and 155.4 °C, respectively. Their melting process occurred in a smaller temperature range and ended at 167.7 °C for the Conall Health A and at 168.0 °C for the Conall Health B/C aprons. The enthalpies of these transitions were also significantly higher, with values of −94 J/g and −161 J/g, respectively. The melting of the Prion Guard suit occurred in the narrowest temperature range, from 156.9 °C to 164.3 °C, with an enthalpy of −44 J/g. The lowest enthalpy of −22 J/g for the thermal degradation was observed for the cotton apron. Its thermal degradation started at the significantly higher temperature of 249.3 °C and ended at 260.4 °C.

The differences in melting of all tested polypropylene fabrics can be ascribed to the different chain structures, their particle sizes, and/or additives used. These factors can change the thermal degradation process as well as the diffusion mechanism of the degradation products, even when the same heating rate is applied [32].

Figure 4 shows the heating DSC curves of the cotton apron in differentiated atmospheres, including an oxygen-free atmosphere, an atmosphere containing 21% oxygen, and one containing 100% oxygen. In the oxygen-free atmosphere, two endothermic peaks are observed, with the first at a temperature range of 250.8–257.9 °C and the second at 333.0–369.7 °C. Both endothermic and exothermic effects are observed in the atmospheres containing 21% and 100% oxygen. The endothermic effects in both atmospheres occurred in a similar temperature range, 249.0–261.4 °C for 21% oxygen and 250.7–257.3 °C for 100% oxygen. The exothermic peaks were observed at higher temperatures with higher enthalpy values of 99 J/g and 301 J/g, respectively. 

### 3.4. Gross Heat of Combustion (Calorific Value)

The gross heat of combustion is determined under an oxygen atmosphere (100% oxygen). The obtained results are therefore not comparable with those of the cone calorimeter, due to the fact that cone calorimeter tests are conducted in an air-flow atmosphere and the combustion takes place under ambient air. 

However, the gross heat of combustion is an important parameter in terms of calculating the fire load density, which is a valuable parameter in fire risk assessment [33]. Table 4 shows the results of determining the gross heat of combustion of the studied fabrics.

The cotton apron has the lowest gross heat of combustion, 19.402 MJ/kg, but its use does not guarantee protection against COVID-19. For the disposable apron, the gross heat of combustion value is 35.375 MJ/kg. The highest values were obtained for the Prion Guard suit, 44.962 MJ/kg, the Conall Health B/C apron, 44.377 MJ/kg, and the Conall Health A apron, 43.981 MJ/kg.

The fire hazard increases with the increasing heat of combustion. The gross heat of combustion of polypropylene fabrics is significantly higher than the cotton apron and may reach up to 45.80 MJ/kg [34], which indicates that polypropylene fabrics provide lower fire safety than the cotton apron. However, the disposable apron exhibits a lower value of gross heat of combustion in comparison to other polypropylene fabrics. This may be due to the content of inorganic fillers, which reduce the heat of combustion [35].

### 3.5. Tensile Properties, Abrasion Resistance, and Electrostatic Discharge Properties

The averaged results of the tensile strength, abrasion resistance, and antistatic properties tests of the studied fabrics are presented in Table 5.

The tensile strength tests, performed to determine the strength and elasticity of the studied fabrics, revealed that the disposable apron, as well as the Conall Health A and B/C aprons, have lower breaking forces and higher corresponding elongations. The lowest maximum force (52.0 N) was observed for the disposable apron. Higher values of 87.2 N and 91.0 N were obtained for the Conall Health A and B/C aprons, respectively. For all specimens, ruptures after reaching the maximum force were observed, with the lowest values for the disposable (1.5%), Conall Health B/C (1.7%), and Conall Health A (2.4%) aprons. A slightly higher elongation at rupture of 5.3% was observed for the Prion Guard suit, with a simultaneous increase in the maximum force (274.5 N) and a decrease in the elongation at maximum force (59.0%) (when comparing to the aforementioned aprons). The highest breaking force (592.1 N) was achieved for the cotton apron, with the lowest corresponding elongation (7.9%) and the highest elongation at rupture (17.3%) among all samples.

The abrasion resistance tests mapping the abrasive wear under controlled conditions exhibited that the Prion Guard suit did not show any changes after exposure to 40,000 abrasion cycles. Among other samples, the cotton apron exhibited the highest abrasion resistance, even though it is reusable, unlike the Prion Guard suit, which is disposable. More than 78% lower abrasion resistance was observed for the Conall Health A, the disposable, and the Conall Health B/C aprons in comparison to the cotton apron, with the latter exhibiting the lowest abrasion resistance among all samples.

The greatest ability to discharge electrostatic charges, expressed by the half decay time of charges, was observed for the Prion Guard suit (0.01 s) and the cotton apron (1 s). These two materials can be classified as electrostatic dissipative materials according to EN 1149-5:2018 Protective clothing—Electrostatic properties—Part 5: Material performance and design requirements [36]. Inferior electrostatic discharge properties were achieved for the disposable, Conall Health A, and B/C aprons, with the former having the longest half decay time of charges (20 s).

## 4. Conclusions

Numerous fires in hospitals during the COVID-19 pandemic drew attention to fire safety in hospitals. Intensive oxygen use may result in an oxygen-enriched atmosphere. The conducted research allowed for the assessment of the fire safety of selected personal protective equipment used by healthcare workers dealing with confirmed COVID-19 patients.

The cone calorimeter test revealed that the cotton apron ignites the fastest (after 10 s), while other studied polypropylene fabrics are more difficult to ignite. The flaming combustion of polypropylene PPE occurred between 42 s and 66 s, with the best results of 60 s and 66 s achieved for the disposable apron and B/C aprons, respectively. Nevertheless, despite the delayed ignition, the pHRR values of the disposable apron (62.70 kW/m^2^) and the Prion Guard suit (61.57 kW/m^2^) exhibited similarly high values to the cotton apron (62.81 kW/m^2^). Moreover, high pHRR values contributed to high THR values of 1.1 MJ/m^2^, 1.0 MJ/m^2^, and 0.9 MJ/m^2^ for the cotton apron, Prion Guard suit, and the disposable apron, respectively. Significantly lower values were observed for the Conall Health A (0.3 MJ/m^2^) and B/C (0.2 MJ/m^2^) aprons. It is worth noting that, despite their better fire behavior under a simulated conditions of stage I of fire development, these two aprons emit the highest amounts of CO—687.51 kg/kg and 700.43 kg/kg, respectively.

The thermal analysis showed that all polypropylene fabrics started to melt at about 100 °C lower temperature comparing to the cotton apron, also exhibiting the lowest enthalpy of transition (−22 J/g). Moreover, the average gross heat of combustion of the cotton apron was the lowest (19.402 MJ/kg). The obtained gross heat of combustion values under a 100% oxygen atmosphere of all tested polypropylene PPE were significantly higher relative to the cotton apron. The highest values were obtained for the Conall Health A (43,981 MJ/kg) and B/C (44,377 MJ/kg) aprons and the Prion Guard suit (44,962 MJ/kg). 

The tensile strength and abrasion resistance test results for the cotton apron and the Prion Guard suit indicate that these two are the most durable. Their half-decay time of charges of 1 s and 0.01 s, respectively, exhibit their low electrification capability. The maximum force values obtained from the tensile strength test of the disposable (52.0 N), Conall Health A (87.2 N), and B/C (91.0 N) aprons were much lower relative to the Prion Guard suit (592.1 N) and the cotton apron (274.1 N).

In conclusion, all the tested materials utilized for personal protective equipment production do not provide sufficient fire safety under oxygen-enriched atmosphere conditions for healthcare workers and patients. Reducing the flammability of materials will improve fire safety in hospitals.

The flammability of PPE made from polypropylene fabrics can be improved using flame retardants in the form of finishes or back-coatings. However, the best results are achieved when retardants are introduced directly into the fiber. Several groups of flame retardants, including phosphorus and silicon containing groups, inorganic hydroxides, and nanoclays, are proposed to disrupt the combustion process in order to improve the fire safety of polypropylene [37].

The durability of PPE is one of the key mitigants to the risk of infection for healthcare workers. Thus, mechanical properties including tensile strength and abrasion resistance should be considered [29]. An increase in durability can be achieved by the application of protective layers on the fabric surface, which improves the hardness and abrasion resistance of fabrics [38].

## Figures and Tables

**Figure 1 ijerph-19-11453-f001:**
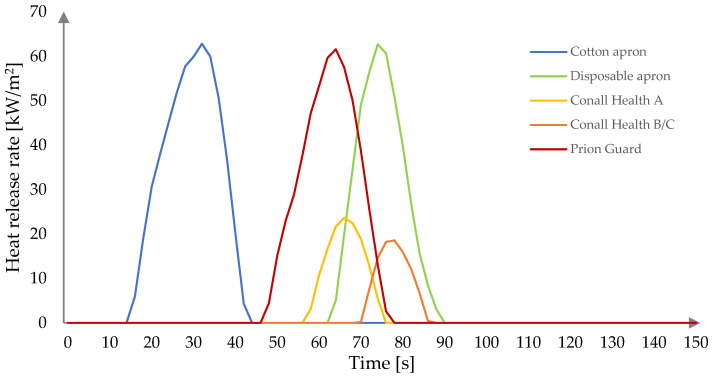
Heat release rate of studied fabrics at heat flux of 25 kW/m^2^.

**Figure 2 ijerph-19-11453-f002:**
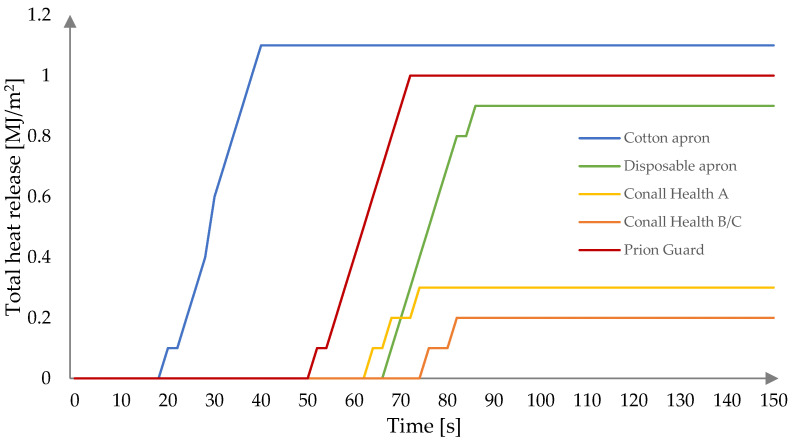
Total heat release of studied fabrics at heat flux of 25 kW/m^2^.

**Figure 3 ijerph-19-11453-f003:**
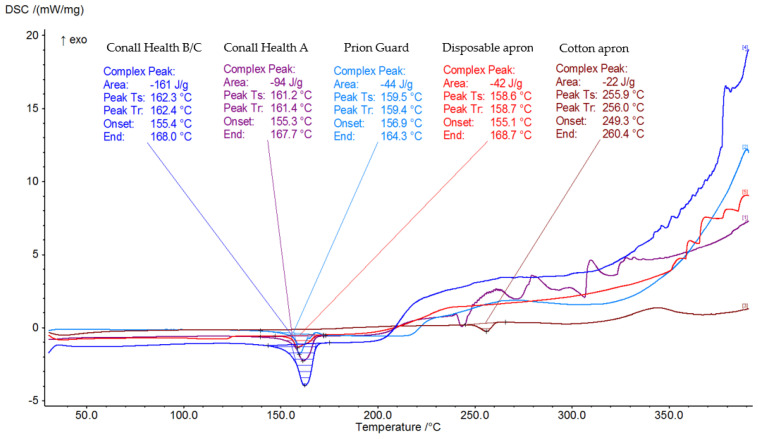
Differential scanning calorimetry curves of all studied fabrics in a nitrogen atmosphere.

**Figure 4 ijerph-19-11453-f004:**
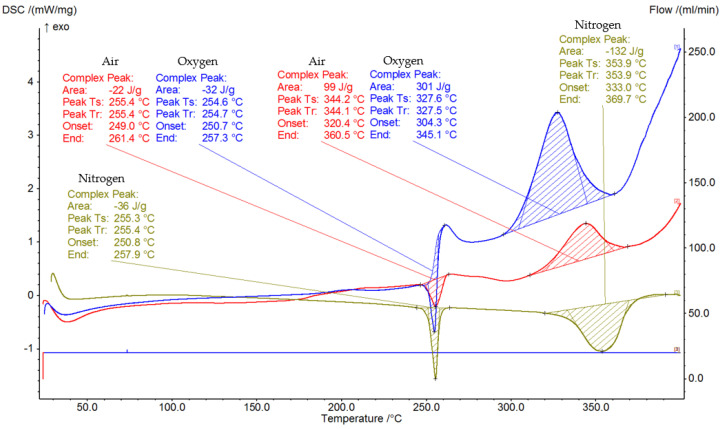
Differential scanning calorimetry curves of cotton fabric in an oxygen-free, 21% oxygen (air), and 100% oxygen (pure oxygen) atmosphere.

**Table 1 ijerph-19-11453-t001:** Composition of fabrics.

Number	Name	Abbreviation	Material	Mass Per Unit Area [g/cm^2^]	Color
1	Cotton apron	CO	Cotton (100%)	0.0206	White
2	Disposable apron	DISP	Non-woven polypropylene fabric with a basis weight of 20 g/m^2^	0.0061	White
3	Conall Health A brand apron	CHA	Non-woven polypropylene fabric with a basis weight of 35 g/m^2^	0.0037	Blue
4	Conall Health B/C brand apron	CHB	Non-woven polypropylene fabric with a basis weight of 35 g/m^2^, with an additional protective inner layer against microbiological agents	0.0036	Blue outer layer and white internal layer
5	Prion Guard suit	PG	Non-woven polypropylene fabric	0.0075	Blue

Note: Material information is taken from the label material description.

**Table 2 ijerph-19-11453-t002:** The LOI values of fabrics.

Fabric	LOI (vol %)	Classification
CO	17.17	Combustible
DISP	17.39	Combustible
CH/A	23.37	Self-extinguishing
CH/B	23.51	Self-extinguishing
PG	24.51	Self-extinguishing

**Table 3 ijerph-19-11453-t003:** Cone calorimeter test results of studied fabrics.

Sample	CO	DISP	CH/A	CH/B	PG
TTI (s)	10	60	50	66	42
pHRR (kW/m^2^)	62.81	62.70	23.65	18.56	61.57
t_pHRR_ (s)	32	74	66	78	64
THR (MJ/m^2^)	1.1	0.9	0.3	0.2	1.0
MARHE (kW/m^2^)	26.40	9.90	3.59	2.19	13.75
Mean CO yield (kg/kg)	132.82	562.46	687.51	700.43	332.65
Mean CO_2_ yield (kg/kg)	40.08	253.80	175.32	173.25	88.47
Residue (%)	32.24	54.82	37.29	50.69	22.68

Abbreviations: TTI—Time To Ignition; pHRR—peak Heat Release Rate; t_pHRR_—time to pHRR; THR—Total Heat Release; MARHE—Minimum Average Rate of Heat Emission.

**Table 4 ijerph-19-11453-t004:** The values of gross heat of combustion of fabrics.

Fabric	Series 1 (MJ/kg)	Series 2 (MJ/kg)	Average Gross Heat of Combustion (MJ/kg)
CO	19.343	19.460	19.402
DISP	35.227	35.523	35.375
CH/A	43.648	44.313	43.981
CH/B	44.496	44.258	44.377
PG	45.919	44.004	44.962

**Table 5 ijerph-19-11453-t005:** Results of the tensile, abrasion resistance, and electrostatic properties tests.

	Tensile Properties			Abrasion Resistance	Electrostatic Discharge Properties
Fabric	Maximum Force (N)	Elongation at Maximum Force (%)	Elongation at Rupture (%)	Number of Cycles	Half Decay Time of Charges t_50_ (s)
CO	592.1 ± 8.8	7.9 ± 0.4	17.3	38,000	1
DISP	52.0 ± 10.2	39.0 ± 1.2	1.5	2000	20
CH/A	87.2 ± 9.2	59.0 ± 1.5	2.4	8000	16
CH/B	91.0 ± 10.3	65.0 ± 1.0	1.7	10,000	15
PG	274.5 ± 6.1	19.1 ± 1.1	5.3	40,000 ^1^	0.01

^1^ No damage to the specimen surface was observed.

## Data Availability

Not applicable.

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
