# Peer review of "Analysis of the Flammability and the Mechanical and Electrostatic Discharge Properties of Selected Personal Protective Equipment Used in Oxygen-Enriched Atmosphere in a State of Epidemic Emergency"

_ijerph, 2022, doi:10.3390/ijerph191811453_

Round 1
Reviewer 1 Report
General remarks:
General 1: The introduction claims that during the COVID-19 pandemic, the oxygen enrichment of the atmosphere has led to an increased number of fire-related accidents. This statement is supported by literature citations from India. This statement is not generally valid, so it has to be put into perspective. Inquiries with the company fire brigade in a hospital with more than 1500 beds showed that the increased use of oxygen did not result in an increased number of fire incidents. In my opinion, it is therefore essential that this statement is put into perspective in the introduction.
General 2: The entire work gives the impression of a test report on 5 different materials than a scientific work. In a scientific paper, I would expect parameters to be varied and effects studied.
Nevertheless, the results are interesting and the work should be published after a thorough revision.
Detailed suggestions for improvement
Section 1
It should be mentioned that in this work actually used materials or textiles are studied. There are a number of studies on the fire behavior of polypropylene (PP) and also how the fire behavior of PP could be improved with flame retardants. One example is:
Section 2.1
The material description is to be greatly improved.
For textiles, the specification of the mass per unit area in g/cm² is useful for the description; the density in g/cm³ is of limited use for sheet-like samples, especially if the thickness is not specified (which is difficult for textiles).
For Cone Calorimeter investigations, the color of the material is essential and should therefore be specified.
For easier readability of this work, material abbreviations should be found, which are used in Tables 2 to 5. It is tedious when reading to look up what e.g. material 3 was.
Section 3
In tables 2 to 5, a short designation for the material (instead of 1 to 5) should be used instead of the material number.
The number of each test should be indicated, or at the beginning that only one test was performed at a time. In the case of the LOI investigation, this is not necessary because there is some certainty in the results due to the staged investigation.
The results of fire tests have a range of variation and therefore the number of tests carried out should be stated for evaluation.
Section 3.2
The heading should not be Fire Behavior but 'Material behavior under thermal irradiation' or similar.
As mentioned above the number of each test should be indicated, or at the beginning that only one test was performed at a time. In the case of cone calorimeter tests, three tests each are necessary for a standard-compliant classification in accordance with ISO 5660-1.
Figure 2 is identical to Figure 1, but with a different (not useful) timeline and should therefore be deleted. Figure 4 is identical to Figure 3, but with a different (not useful) timeline and should therefore be deleted.
You might want to mention that the total duration of the experiment was 1200 seconds.
The material 4 has an inner protective layer against microbiological agents. In the total heat release (THR), this can be seen very clearly as a step in the THR diagram (old Fig. 3). This should mentioned in the paper. In this context, it is surprising that this layer is not visible in the curve for the heat release rate (Fig. 1).
Section 3.3
Line 198: The effect shows the melting of the semi-cristalline material. I would not use ‘decomposition’ which means breaking of bonds. Such effects are visible at temperatures above
200 °C.
Section 3.4
It should be mentioned: The gross heat of combustion is determined under an oxygen atmosphere (100% oxygen). These results are therefore not comparable with those of the Cone calorimeter, because there the combustion takes place under ambient air.
Section 4
The first paragraph of the Conclusion should be qualified according to my point General 1.
Author Response
Review 1
Thank you for giving us the opportunity to submit a revised draft of the manuscript titled ‘Analysis of the flammability, mechanical and electrostatic discharge properties of selected personal protective equipment used in oxygen-enriched atmosphere in the state of epidemic emergency’ to International Journal of Environmental Research and Public Health. We appreciate the time and effort that you have dedicated to providing your valuable feedback. We are grateful for the insightful comments on our paper. We have been able to incorporate changes to reflect all the suggestions. Here is a point-by-point response to the mentioned comments and concerns.
General remarks:
General 1: The introduction claims that during the COVID-19 pandemic, the oxygen enrichment of the atmosphere has led to an increased number of fire-related accidents. This statement is supported by literature citations from India. This statement is not generally valid, so it has to be put into perspective. Inquiries with the company fire brigade in a hospital with more than 1500 beds showed that the increased use of oxygen did not result in an increased number of fire incidents. In my opinion, it is therefore essential that this statement is put into perspective in the introduction.
Response: The statement had been updated. More information of the fire accidents according to the JRC report had been added.
General 2: The entire work gives the impression of a test report on 5 different materials than a scientific work. In a scientific paper, I would expect parameters to be varied and effects studied.
Response: The analysis of results including of the analysis of the gross heat of combustion values and analysis of the heat release rate and total heat release had been added in order to extend the discussion.
Detailed suggestions for improvement
Section 1
- It should be mentioned that in this work actually used materials or textiles are studied. There are a number of studies on the fire behavior of polypropylene (PP) and also how the fire behavior of PP could be improved with flame retardants.
Response: The fire behavior of polypropylene fabrics was mentioned in the introduction. The information of the fire behavior of PP containing flame retardants has been added in the form of a recommendation for the improvement of flammability of PPE.
Section 2.1
- The material description is to be greatly improved. For textiles, the specification of the mass per unit area in g/cm² is useful for the description; the density in g/cm³ is of limited use for sheet-like samples, especially if the thickness is not specified (which is difficult for textiles).
Response: As manufacturers do not provide detailed information on the composition of the fabrics, some characteristics are not known unfortunately. However, the material description had been supplemented with data on the mentioned mass per unit area in g/cm2.
- For Cone Calorimeter investigations, the color of the material is essential and should therefore be specified.
Response: Because the principle of cone calorimetry is based on the measurement of the decreasing oxygen concentration in the combustion gases of a sample subjected to a given heat flux, the color of the materials does not affect the measurement results.
- For easier readability of this work, material abbreviations should be found, which are used in Tables 2 to 5.
Response: Material abbreviations were introduced in Table 1 and are further used in Tables 2-5.
Section 3
- In tables 2 to 5, a short designation for the material (instead of 1 to 5) should be used instead of the material number.
Response: Material abbreviations were introduced in Table 1 and are further used in Tables 2-5.
- The number of each test should be indicated, or at the beginning that only one test was performed at a time. In the case of the LOI investigation, this is not necessary because there is some certainty in the results due to the staged investigation. The results of fire tests have a range of variation and therefore the number of tests carried out should be stated for evaluation.
Response: The information of the number of conducted measurements has been added in subsections 2.2.1 - 2.2.7.
Section 3.2
- The heading should not be Fire Behavior but 'Material behavior under thermal irradiation' or similar.
Response: The heading ‘Fire behavior’ is related to the fact that the cone calorimeter tests simulate the behavior of materials in conditions of stage I of fire development (at a radiant heat flux density of 25 kW/m2) for interior finishing materials and stage II of fire development (at a radiant heat flux density of 50 kW/m2) for fire partition elements.
- As mentioned above the number of each test should be indicated, or at the beginning that only one test was performed at a time. In the case of cone calorimeter tests, three tests each are necessary for a standard-compliant classification in accordance with ISO 5660-1.
Response: The information of the number of conducted measurements had been added in subsections 2.2.1 - 2.2.7. The cone calorimeter tests were performed in three repetitions.
- Figure 2 is identical to Figure 1, but with a different (not useful) timeline and should therefore be deleted. Figure 4 is identical to Figure 3, but with a different (not useful) timeline and should therefore be deleted. You might want to mention that the total duration of the experiment was 1200 seconds.
Response: As suggested, Figures 1 and 3 has been deleted. The information of the total duration of the experiment has been added.
- The material 4 has an inner protective layer against microbiological agents. In the total heat release (THR), this can be seen very clearly as a step in the THR diagram (old Fig. 3). This should be mentioned in the paper. In this context, it is surprising that this layer is not visible in the curve for the heat release rate (Fig. 1).
Response: The information about THR curves has been added. We also added comment that this layer is indeed not visible in the curve for the heat release rate (Fig. 1).
Section 3.3
- Line 198: The effect shows the melting of the semi-cristalline material. I would not use ‘decomposition’ which means breaking of bonds. Such effects are visible at temperatures above 200 °C.
Response: As correctly suggested, the term ‘decomposition’ has been changed to melting.
Section 3.4
- It should be mentioned: The gross heat of combustion is determined under an oxygen atmosphere (100% oxygen). These results are therefore not comparable with those of the Cone calorimeter, because there the combustion takes place under ambient air.
Response: The information of the conditions and incomparability has been added, as well as the information of the possibility to use the gross heat of combustion in the calculation of the fire load density.
Section 4
- The first paragraph of the Conclusion should be qualified according to my point General 1.
Response: The first paragraph of the Conclusions has been changed according to the given suggestions.

Reviewer 2 Report
1. The abstract needs to reflect the significant findings (quantitative) of this research. Key outcomes from the result and discussion shall elaborate in a concise manner in the abstract, before coming into the conclusion.
2. Too many keywords. The important keywords shall limit to Five.
3. Intense Proofreading and proof editing of English is required.
4. In section 2.1, Table 1. Composition of fabrics, please provide sources of reference for each type of apron and protective suit listed.
5. In Section 2.2. Methods, each experiment in paragraph should clearly separated into appropriate sub-headings. For example, 2.2.1. Limiting oxygen index (LOI), 2.2.2. Cone calorimeter test, 2.2.3. Differential scanning calorimeter, 2.2.4. Oxygen bomb calorimeter, 2.2.5. Instron tensile test, etc.
6. The similarity of this article is 21%. Please revise the article to similarity below 20%.
7. Line 139-140, Please state the actual quantitative figure to indicate how many numerous repetitions of all measurements were conducted.
8. Please show photos of LOI and cone calorimeter test results of all specimens in section 3.1 and 3.2.
9. Please remove Figure 2. It seems repetitive.
10. For figure 1, Please provide justification why HRR of Conall Health B/C is the lowest compared to the rest of specimens.
11. Please remove Figure 4, it is repetitive.
12. Please add Pyrolysis gas chromatography mass spectrometry test for all specimens to reveal the involatile compounds released.
13. In line 209-210, please provide the possible fact of chain structures, particle sizes, additives used etc. for all polypropylene fabrics tested.
14. Please provide legends for all tested specimens in Figure 6.
15. Please provide qualitative explanation for results obtained in Section 3.4.
16. Please highlight the quantitative outcome in Section 4. Conclusions.
16. In the last clause of Section 4. Conclusion (line 285-288), please provide specific recommendations for the thermal and mechanical properties improvements of PPE.
17. Please check the referencing style and format. Some of them are inconsistence.
Author Response
Review 2
Thank you for giving us the opportunity to submit a revised draft of the manuscript titled ‘Analysis of the flammability, mechanical and electrostatic discharge properties of selected personal protective equipment used in oxygen-enriched atmosphere in the state of epidemic emergency’ to International Journal of Environmental Research and Public Health. We appreciate the time and effort that you have dedicated to providing your valuable feedback on this manuscript. We are grateful for the insightful comments on our paper. We have been able to incorporate changes to reflect all the suggestions. Here is a point-by-point response to the mentioned comments and concerns.
- 1. The abstract needs to reflect the significant findings (quantitative) of this research. Key outcomes from the result and discussion shall elaborate in a concise manner in the abstract, before coming into the conclusion.
Response: The significant quantitative findings of the research were included and specified in the abstract.
- 2. Too many keywords. The important keywords shall limit to Five.
Response: The number of keywords had been limited to five.
- 3. Intense Proofreading and proof editing of English is required.
Response: We must agree, the English language in the original manuscript was indeed clumsy at some points. Numerous linguistic corrections have been made to correct the text. Professional proofreading has been performed on this manuscript.
- 4. In section 2.1, Table 1. Composition of fabrics, please provide sources of reference for each type of apron and protective suit listed.
Response: The fabrics that constitute the subject of this research come from several different companies. Manufacturers (Rates, Pinaldi, Conall Health, and Priontex) do not provide detailed information of individual products, and they are not widely available. It is difficult and challenging to find references to specific products directly on their websites (due to the selling by wholesale by the manufacturers, except company ‘Rates (Poland)’). The distributors command information found on the product labels, and this has been included in Table 1.
- 5. In Section 2.2. Methods, each experiment in paragraph should clearly separated into appropriate sub-headings. For example, 2.2.1. Limiting oxygen index (LOI), 2.2.2. Cone calorimeter test, 2.2.3. Differential scanning calorimeter, 2.2.4. Oxygen bomb calorimeter, 2.2.5. Instron tensile test, etc.
Response: As suggested, each experiment methodology has been separated into subsections.
- 6. The similarity of this article is 21%. Please revise the article to similarity below 20%.
Response: We have revised the manuscript extensively, now we hope that after this major revision the similarity is at an acceptable level.
- 7. Line 139-140, Please state the actual quantitative figure to indicate how many numerous repetitions of all measurements were conducted.
Response: The information of the number of conducted measurements has been added in subsections 2.2.1 - 2.2.7.
- 8. Please show photos of LOI and cone calorimeter test results of all specimens in section 3.1 and 3.2.
Response: For the cone calorimeter study, due to the fact that the residues were not well enough visible, the pictures were not taken at that point. In order to provide the requested photos, we would have to perform the studies once again. Unfortunately, due to the current malfunction and the necessary of repair and service of the cone calorimeter, it is not possible to repeat the measurements to take pictures of residues.
- Please remove Figure 2. It seems repetitive. Please remove Figure 4, it is repetitive.
Response: Since after the time of 100 s the HRR and THR curves for all tested fabrics were the same and the values of HRR and THR were 0, the Figures with the longer timeline (1200 s) have been deleted and this indeed allows to present the results in the best/most visible way. However, the information of the total duration of the experiment has been added.
- For figure 1, Please provide justification why HRR of Conall Health B/C is the lowest compared to the rest of specimens.
Response: As suggested, the justification has been provided in the revised manuscript.
- Please add Pyrolysis gas chromatography mass spectrometry test for all specimens to reveal the involatile compounds released.
Response: It is a valuable comment, and GC-MS is a powerful tool to study products of combustion and/or pyrolysis. Please note that the fabrics that are the subject of this research article were investigated using seven different test methods involving the determination of thermal and mechanical properties and indeed further and more advanced tests are planned for a subsequent article. The authors are grateful for this valuable comment. The pyrolysis gas chromatography mass spectrometry will be probably included in the next research paper. In addition, while studies of evolved volatile species by GC-MS is quite straightforward, study of involatile compounds is a more serious research challenge. Please also note that studies of such type can be found in literature (for example: High resolution gas chromatographic–mass spectrometric analysis of polyethylene and polypropylene thermal cracking products, Journal of Analytical and Applied Pyrolysis, Volume 78, Issue 2, 2007, Pages 387-399). We hope that at this point this response is sufficient.
- In line 209-210, please provide the possible fact of chain structures, particle sizes, additives used etc. for all polypropylene fabrics tested.
Response: This is indeed a very important piece of information. Nevertheless, as manufacturers do not provide detailed information on the composition of the fabrics, including chain structures, particle sizes and additives used, some characteristics remain not known, unfortunately. However, please note that the materials description has been supplemented with data on the mass per unit area in g/cm2 useful for the description of fabrics.
- Please provide legends for all tested specimens in Figure 6.
Response: As mentioned in the description, Figure 6 (as numbered in the original manuscript, in the revised text – Figure 4) refers to the differential scanning calorimetry curves of cotton fabric in an oxygen-free, 21% oxygen (air) and 100% oxygen (pure oxygen) atmosphere. Since the behavior of the only one, selected material (the cotton apron) is presented, the authors did not put that into the legend. However, please kindly note that the legend contains the information of the atmosphere under which the measurements were conducted.
- Please provide qualitative explanation for results obtained in Section 3.4.
Response: According to this suggestion, qualitative explanation for the results of the gross heat of combustion test has been added in the revised manuscript.
- Please highlight the quantitative outcome in Section 4. Conclusions.
Response: The quantitative outcome has been included in the Section 4 of the revised manuscript.
- In the last clause of Section 4. Conclusion (line 285-288), please provide specific recommendations for the thermal and mechanical properties improvements of PPE.
Response: Recommendations for the improvement of flammability and mechanical properties of PPE have been added.
- Please check the referencing style and format. Some of them are inconsistence.
Response: The references have been revised and changed/corrected to keep one particular style.

Round 2
Reviewer 1 Report
Review of the revised version of the paper
Analysis of the flammability, mechanical and electrostatic discharge properties of selected personal protective equipment used in oxygen-enriched atmosphere in the state of epidemic emergency
Authors: Adriana Dowbysz *, Bożena Kukfisz *, Dorota Siuta, Mariola Samsonowicz, Andrzej Maranda, Wojciech Wróblewski
The paper was intensively revised by the authors and not all my suggestions were taken into account, but this is not discussed further.
Important in the version now presented are the following 2 points:
In Table 1, the density is now given in g/cm³ and the mass per unit area in g/cm². If you divide the surface-related mass by the density, you get the material thickness. If you do this for the DISP material, you get a material thickness of 0.0003 cm, which seems very thin to me. I suspect there is a mistake in the units, which should be urgently improved.
My comments on point 2 of the authors' reply
- For Cone Calorimeter investigations, the color of the material is essential and should therefore be specified.
Response: Because the principle of cone calorimetry is based on the measurement of the decreasing oxygen concentration in the combustion gases of a sample subjected to a given heat flux, the color of the materials does not affect the measurement results.
Comment of the reviewer on this (response) statement:
There is a mistake in the authors' understanding here:
The material is irradiated with thermal energy and the test specimen is heated by the radiation and the resulting gases are finally ignited with the pilot flame. According to Boltzmann's radiation law, the absorbed radiation energy of the test specimen depends on the absorption coefficient of the test specimen and thus on the color. So it makes a significant difference whether an identical material is examined in white or black color, which is also reflected in the measurement results. Also, the material color is essential and should be indicated, or for example that all samples were white.
If these two points are improved, the work can be published.
Author Response
Review 1 (Round 2)
Thank you for giving us the opportunity to submit a revised draft of the manuscript titled ‘Analysis of the flammability, mechanical and electrostatic discharge properties of selected personal protective equipment used in oxygen-enriched atmosphere in the state of epidemic emergency’ to International Journal of Environmental Research and Public Health. We incorporated changes to both suggestions.
- In Table 1, the density is now given in g/cm³ and the mass per unit area in g/cm². If you divide the surface-related mass by the density, you get the material thickness. If you do this for the DISP material, you get a material thickness of 0.0003 cm, which seems very thin to me. I suspect there is a mistake in the units, which should be urgently improved.
Response: Thank you for your important comment. There is a mistake in the unit. The basis weight (column 3) of non woven polypropylene fabric used by the producer should have been given in g/m2. That information is only given for the disposable (20 g/m2), Conall Health A (35 g/m2) and B/C (35 g/m2) aprons. The mass per unit area [g/cm2] of aprons and a suit were measured by the authors. As a result, the thickness of the fabric cannot be calculated.
- My comments on point 2 of the authors' reply:
- For Cone Calorimeter investigations, the color of the material is essential and should therefore be specified.
Response: Because the principle of cone calorimetry is based on the measurement of the decreasing oxygen concentration in the combustion gases of a sample subjected to a given heat flux, the color of the materials does not affect the measurement results.
Comment of the reviewer on this (response) statement:
There is a mistake in the authors' understanding here:
The material is irradiated with thermal energy and the test specimen is heated by the radiation and the resulting gases are finally ignited with the pilot flame. According to Boltzmann's radiation law, the absorbed radiation energy of the test specimen depends on the absorption coefficient of the test specimen and thus on the color. So it makes a significant difference whether an identical material is examined in white or black color, which is also reflected in the measurement results. Also, the material color is essential and should be indicated, or for example that all samples were white.
Response: Colors of the materials are given in Table 1 (column 6).
